# Gas Metal Arc Welding Modes in Wire Arc Additive Manufacturing of Ti-6Al-4V

**DOI:** 10.3390/ma14092457

**Published:** 2021-05-10

**Authors:** Oleg Panchenko, Dmitry Kurushkin, Fedor Isupov, Anton Naumov, Ivan Kladov, Margarita Surenkova

**Affiliations:** 1Laboratory of Lightweight Materials and Structures, Institute of Machinery, Materials, and Transport, Peter the Great St. Petersburg Polytechnic University, 29 Polytechnicheskaya St., 195251 St. Petersburg, Russia; panchenko_ov@spbstu.ru (O.P.); anton.naumov@spbstu.ru (A.N.); kladov.iv@edu.spbstu.ru (I.K.); rita-surenckova@yandex.ru (M.S.); 2The World-Class Advanced Digital Technologies Research Center, Peter the Great St. Petersburg Polytechnic University, 29 Polytechnicheskaya St., 195251 St. Petersburg, Russia; isupov_fyu@spbstu.ru

**Keywords:** wire arc additive manufacturing, Ti-6Al-4V, gas metal arc welding, metal transfer modes, waveforms, microstructure

## Abstract

In wire arc additive manufacturing of Ti-alloy parts (Ti-WAAM) gas metal arc welding (GMAW) can be applied for complex parts printing. However, due to the specific properties of Ti, GMAW of Ti-alloys is complicated. In this work, three different types of metal transfer modes during Ti-WAAM were investigated: Cold Metal Transfer, controlled short circuiting metal transfer, and self-regulated metal transfer at a direct current with a negative electrode. Metal transfer modes were studied using captured waveform and high-speed video analysis. Using these modes, three walls were manufactured; the geometry preservation stability was estimated and compared using effective wall width calculation, the microstructure was analyzed using optical microscopy. Transfer process data showed that arc wandering depends not only on cathode spot instabilities, but also on anode processing properties. Microstructure analysis showed that each produced wall consists of phases and structures inherent for Ti-WAAM. α-basketweave in the center of and α-colony on the grain boundary of epitaxially grown β-grains were found with heat affected zone bands along the height of the walls, so that the microstructure did not depend on metal transfer dramatically. However, the geometry preservation stability was higher in the wall, produced with controlled short circuiting metal transfer.

## 1. Introduction

In recent years, WAAM of Ti-alloy parts (Ti-WAAM) has been rapidly developed according to the rising demand of cheap high-scale Ti-alloy parts. One of the most required Ti-alloys is Ti-6Al-4V, which is broadly used in the aerospace industry due to its low density, high strength-to-weight ratio, outstanding corrosion resistance, and good performance at an elevated temperature [1,2,3]. It is also used for bio-medical implants owing to its excellent biocompatibility [4].

Generally, Ti-WAAM is performed using either gas tungsten arc welding (GTAW) [5,6,7,8] or plasma arc welding (PAW) [9,10,11]. In GTAW and PAW the welding arc (or plasma) is between the negative tungsten electrode—cathode, and the positive substrate—anode. Under such conditions, the steady arc (or plasma) heats the metal of the substrate and the filler metal at the necessary rate while the heat input is generally controlled by the means of altering the current, the voltage, the wire feed rate, and the travel speed. Once the synergetic relationship of the four mentioned basic parameters is established, the metal transfer during the manufacturing proceeds without issue.

Usually GTAW- and PAW-WAAM parts have no defects, so researchers are concerned with resolving the issues regarding geometric parameters or the microstructure using different post-processing or interpass technologies, namely hot isostatic pressing [12], rolling [13], active interpass cooling [14], ultrasonic impact treatment [15], etc. Rare studies are to be found representing investigation on the influence of current and voltage waveforms on the deposited metal parameters. For instance, Benakis et al. reported that a single Pulsed-GTAW system could be used to manufacture different bead geometries applying low or high frequency pulsing [16]. Nevertheless, the metal transfer during GTAW or PAW is not controlled directly since it depends mainly on the heat input.

Compared to GTAW and PAW, gas metal arc welding (GMAW) has a decisive bene-fit—in GMAW the arc exists between the substrate and the filler wire, which is fed coaxially to the arc. This advantage drastically increases the overall flexibility of WAAM with GMAW, which is why this method is regularly used in WAAM of steels [17] and Al-alloys [18]. Moreover, there is a possibility of modifying the current and voltage waveforms during GMAW so that the metal transfer can be adjusted to specifically influence the deposited metal properties [19]. Owing to the aforementioned benefits, GMAW should be considered as the predominant welding method for Ti-WAAM, as well as for the WAAM of Al-alloy parts or steel parts. In GMAW, the filler metal is transferred by one of the three conventional transfer modes: spray, globular, or short-circuiting [20]. Higher current contributes to the formation of strong electromagnetic forces that influence the electrode wire tip so that spray transfer of small discrete metal drops occurs. Lower current melts the electrode and forms droplets with the size of the electrode wire diameter and globular transfer of those droplets under the influence of gravity occurs. When the wire touches the welding pool surface and creates a short circuit, short-circuiting transfer of molten metal from the tip of the wire occurs; the metal in this case is transferred under the influence of surface tension force. This last mode operates with the lowest currents resulting in the lowest heat input, which makes it highly beneficial for application in WAAM. High frequency digital feedback control power sources allow modification of conventional metal transfer modes, i.e., the modified controlled short-circuiting metal transfer, Cold Metal Transfer (CMT), operates with the reciprocating movement of the filler wire, which makes it possible to control each metal transfer cycle precisely, resulting in even lower heat inputs, than those obtained with the use of the conventional short-circuiting metal transfer mode [21].

However, GMAW of Ti-alloys at relatively low currents (lower than 300 A [22]) has been a subject of discussion over the past decades. The inherent problems for Ti-alloys occur when establishing the arc between the two Ti electrodes i.e., between the substrate and the wire—the arc wandering phenomenon [23] and the excessive spattering due to a cathode spot relocation [22].

Various attempts have been made to stabilize the metal transfer during GMAW of Ti-alloys. Zhang et al. developed a modified active control of pulsed GMAW for Ti-alloy wire metal transfer and significantly increased the metal transfer process stability compared to the conventional-pulsed GMAW [24]. Otani demonstrated that the arc wandering phenomenon can be confined by applying a high-frequency micro-oscillation to the tip of the welding torch [25]. Choudhury et al. [26] researched the stability of the GMAW process during WAAM using surface treated commercially pure Ti wire and reported on a significant process improvement when using wire with an oxidized surface. Many works reported on Ti-GMAW stabilization using laser beam—arc stabilization was defined as the lack of cathode spot motion, the cathode spot position was confined to the laser beam focus spot location [23,27]. Similar results were reported in later works by different researchers [28,29,30,31,32,33]. Sun et al. investigated the droplet transfer behavior in CMT during Ti-alloy welding and found that arc blow with excessive spattering occurs when the welding speed is relatively low [34].

Thus, GMAW is utilized in Ti-WAAM less frequently than GTAW or PAW due to complications regarding the metal transfer of the Ti-wire during manufacturing. Several studies did not report any issues concerning metal transfer during Ti-WAAM using CMT [35,36]. Nevertheless, Pardal et al. demonstrated that the effective wall width that could be produced by the CMT process is insufficient, which is why they applied a low-energy laser beam for arc stabilization and achieved positive results [37]. However, the use of the laser in WAAM increases the equipment cost, consequently increasing production expenses, which is why a simple solution for the Ti-WAAM at low currents is still considered to be a requirement.

The aim of the present work was to investigate and compare three metal transfer modes in Ti-WAAM: CMT mode, controlled short-circuiting (CSC) mode, and a conventional mode at a direct current with a negative electrode (DCEN). The investigation was provided with the acquired current and voltage waveforms and transfer process high-speed video analysis. Moreover, the influence of the transfer mode on the geometrical parameters and the metal properties of the manufactured structures was evaluated. The significance of the present research is highlighted by the opportunity provided to obtain clearer information on the arc wandering phenomenon during GMAW-based Ti-WAAM.

## 2. Materials and Methods

The walls were printed using the WAAM system which was composed of a Motoman MH24 robot (Yaskawa, Kitakyushu, Japan) and an Alpha Q 552 welding power source (EWM Group, Mündersbach, Germany) or a TransPuls Synergetic 3200 welding power source (Fronius, Wels, Austria). Ti-6Al-4V alloy was used as both the filler wire and the initial substrate. The filler wire had a diameter of 1.2 mm and the substrate was 8 mm thick. Pure argon (99.99%) was used as a shielding gas, the metal was deposited inside a non-vacuum sealed box with a controlled argon atmosphere.

Three walls were printed using CMT, CSC, and DCEN metal transfer modes. CMT was performed using Fronius TransPuls Synergetic 3200 power source, CSC and DCEN were performed using EWM Alpha Q 552 power source. The travel speed was 20 cm/min, the wire feed rate was 4.5 m/min and these parameters did not vary between the printed walls in order to compare the metal transfer modes. The average current and average voltage values are shown in Table 1.

The metal transfer process was shot using a Chronos high-speed video camera (Kron Technologies Inc., Burnaby, BC, Canada) and a backlight 40 W LED matrix. The current and voltage waveforms were captured at a frequency of 100 kHz using a GW Instek GDS-71104B oscilloscope and a 300 μOhm shunt when the Fronius TransPuls Synergetic 3200 power source was used. In the case when the EWM Alpha Q 552 power source was used, voltage and current were recorded from the power source output information port at a frequency of 24 kHz. Captured waveforms were processed via Matlab and noises filtered using a median filter with a 15 samples window and a moving average filter with a 5 samples window.

Transverse sections of walls were prepared using the Buehler grinding and polishing system. Grinding was carried out in three steps with a gradual increase in the grit of the SiC sandpaper: p240 was used for 2 min, p800 for 2 min, and p1000 for 3 min. The sand-paper and the sample holder were rotated in controversial directions, the rotation speeds of the sandpaper and the sample holder were 220 rpm and 160 rpm respectively. The load for pressing the samples was 30 N.

Polishing was carried out in four steps with a gradual decrease in the diamond sus-pension grain size: 9 μm and 6 μm suspensions were used with an NX-MET X500 pol-ishing cloth, 3 μm and 1 μm suspensions were used with the NX-MET X220 polishing cloth. Polishing time was 5 min for each suspension, the polishing cloth and the sample holder were rotated in the same direction. The rotation speeds of the polishing cloth and the sample holder were 120 rpm and 160 rpm respectively, the load for pressing the sam-ples was 25 N.

The cross sections were etched using a solution of 96 mL of distilled water, 2 mL of hydrofluoric acid, and 2 mL of nitric acid; the samples were kept in the etchant for 75 s. A Leica DMI 5000M metallographic microscope was used to obtain the microstructure images.

## 3. Results and Discussion

### 3.1. CMT

Three CMT transfer cycle waveforms are represented in Figure 1. The typical transfer cycle starts with high current (150 A) ignition—at 20.8 ms in Figure 1. Simultaneously, the wire is pulled back from the welding pool for approximately 1.5 ms, which is the special feature of the CMT process. After ignition, the arc burns at an average current of 65 A and the wire is fed towards the substrate until short circuiting occurs (at 34 ms) and the metal is transferred via surface tension. During the short circuiting, the current is mostly maintained at the low level of 20 A, the metal transfer cycle duration is 16.7 ms.

Figure 2 represents photos of the discussed cycle. The arc length decreases rapidly during the arc burning phase and no metal droplet is formed at the wire tip, which indicates insufficient heat input into the filler wire. During the ignition phase, the wire is pulled back at a great distance from the welding pool, which results in a significantly high arc length of approximately 4.5 mm, burning at such a length for 3 ms. The cathode spot, which is the brightest spot on the substrate, appears to be stable and is situated right under the wire tip.

The second frame in Figure 2 shows that the burning arc at the ignition start misses the wire tip, which occurs almost in every cycle and can significantly affect the heat input distribution. A closer look at the second frame is shown in Figure 3—it can be noted that the arc misses the wire for approximately 0.8 ms at the ignition start when the wire is pulled back at a maximum height. Such a phenomenon can be explained by the cumulative effect of the low wire tip temperature and the refractory properties of Ti, which results in insufficient anode vaporization. The lack of anode vaporization develops in the diffusive arc burning mode [38] until the wire tip is heated enough to produce an anode spot.

Despite the fact that the processes occurring at the anode are considered much less often than the processes occurring at the cathode [39], the discovered effect tends to play a crucial role in the arc burning stability during CMT. On the other hand, the observed metal transfer data exhibits a quite stable cathode spot without any wandering, which does not correlate with that presented in other study results [37].

Rare spattering can be observed during the CMT process (Figure 4), spatters with an approximate size of 0.4 mm can be uncommonly repulsed from the welding pool at the ignition start. The Lorentz-type pinch forces caused by the high current flow through the curved welding pool surface to the cathode spot after the wire is forcefully pulled back can result in such a repulsion [22].

### 3.2. CSC

Three CSC transfer cycle waveforms are represented in Figure 5. The typical transfer cycle starts with high current (170 A) ignition—at 51.5 ms in Figure 5. After the 5 ms ignition the arc burns at an average current of 120 A for 44 ms. Next, the wire touches the welding pool and the short-circuiting phase starts, at which the current is maintained at 130 A for an average of 8 ms. At the end of the short-circuiting phase, when the molten metal bridge between the wire and the welding pool starts shrinking, the current is lowered to the minimum possible to minimize the Lorentz pinch force, which affects the bridge and can cause spattering. The metal transfer cycle duration is 52.7 ms.

Figure 6 displays photos of the CSC transfer cycle. The wire in the case of the CSC transfer is not pulled back, thus the arc length is self-regulated. The self-regulated arc length is significantly lower than that of the CMT during the arc ignition phase and differs from 1 to 2 mm. Compared to CMT, a longer arc burning phase results in a big metal droplet on the wire tip formation during CSC. Moreover, the higher heat input during the short-circuiting phase in the case of CSC results in higher wire tip temperatures at the arc ignition phase, which is beneficial in terms of anode spot formation and hence arc burning stabilization.

Figure 7 represents rare yet significant cathode spot wandering during CSC. The cathode spot on the surface of the Ti-alloy workpiece has an inherent habit to change its position erratically—this phenomenon is generally associated with the specific physical properties of Ti i.e., boiling temperature, thermal conductivity, work function, and emis-sivity [23]. However, the cathode spot wandering rarely occurs right before the short-circuiting phase, thus resulting merely in a minor influence on the heat input distribution.

### 3.3. DCEN

Three DCEN transfer cycle waveforms are represented in Figure 8. The typical transfer cycle starts with high current (280 A) ignition—at 27.2 ms in Figure 8. Then the current is rapidly lowered to an average level of 150 A. Next short circuiting occurs, during which the current linearly and rapidly rises because of the increasing resistance of the welding pool, which is raised by the rising short-circuiting current resulting in raising the temperature. The extremely raised short circuiting current does not drop when the wetting bridge is terminated and the arc starts burning at the current, at which the short circuiting ended. This process is completely self-regulated. The metal transfer cycle duration is 29.0 ms.

Figure 9 displays photos of the DCEN transfer cycle. The significant difference between DCEN and the previously discussed transfer modes is that in this case the cathode spot is located at the wire tip. If the wire tip section is not larger than the wire section (a droplet is not formed), then the cathode spot is strictly located on the wire tip, which is highly beneficial for arc burning stability. The arc length is also low here as in the case of the CSC transfer, which is beneficial for the heat input distribution.

However, as shown in Figure 10, if a considerably large metal droplet is formed at the wire tip, the cathode spot can wander significantly on the surface of this droplet. Moreover, spattering frequently occurs at the end of the short-circuiting phase (Figure 11) due to the extremely high current, resulting in strong pinch forces. These are both disadvantages of self-regulation during the DCEN transfer mode.

Full high-speed videos of the CMT, CSC, and DCEN metal transfer processes are available as the Appendix A in the provided dataset [40].

### 3.4. Microstructure

Figure 12 displays cross-sections of the walls, produced using the metal transfer modes discussed above. The CMT and CSC walls were printed in six layers; however, the geometry preservation inconsistency during the DCEN wall manufacturing showed that there is no possibility in DCEN walls printing using the same or similar travel speeds and wire feed rates as in the case of CMT and CSC samples; therefore only four layers of the DCEN wall were printed in order to obtain comparable results.

The effective wall width considerably differs between walls (Figure 12). It is possible to obtain a wall with an effective width of 3.3 mm out of a 5.3 mm CMT wall, while a 5.6 mm wall can be obtained out of a 7.3 mm CSC wall. Efficiency can be assessed through the ratio of the effective wall width to the maximum wall width; thus, the efficiency is 0.62, 0.76, and 0.56 for the CMT, CSC, and DCEN walls, respectively. Similar efficiency of the CMT printed walls was demonstrated in another researchers’ work [26,37].

Here, the wall width efficiency is influenced by the wall waviness, which depends on the heat input distribution. Heat input distribution varies during GMAW due to arc wandering, and its effect was greater in the CMT process as a result of the extended arc length. The DCEN wall inconsistency is a consequence of the higher heat input into the anode [20], which in this case was the wall. Relatively high CSC wall efficiency is the result of the low arc length and sufficient heat input, produced during the arc burning phase of the metal transfer cycle.

Each wall microstructure in Figure 12 show that the prior β grains grow epitaxially in the build direction with dimensions varying from 100 µm to 7000 µm; the same β-grain morphology was found as is usual in WAAM-produced structures by other researchers [41,42,43]. The average coarse β-grain height is 3000 µm in the CMT and CSC walls, while the average coarse β-grain height in the DCEN wall is 4300 µm. A similar difference can be observed in the coarse β-grain width, which is 900 µm in the CMT and CSC and 1200 µm in the DCEN walls. This variation in primary grain size can be explained by the significant difference in the heat input into the substrate between GMAW EN and GMAW EP [20], which resulted in the difference between the cooling rates and hence the primary grain size.

The microstructure images (Figure 13a,c,e) represent an α-basketweave structure within the columnar prior β-grains. Such a microstructure was formed due to repeated rapid heating and cooling that takes place in the WAAM process [43]. The α-lath thickness is about 2 μm and does not differ between the three investigated wall microstructures.

Grain boundary α forms at the prior β-grain boundary and there is a preferred orientation between the grain boundary α and the primary β-grain [41,42], which leads to the formation of lamellar α-colonies with an average α-lath thickness of about 1 μm (Figure 13b,d,f). Vazquez et al. [36] reported that this type of microstructure with the α-colonies along primary β-grains exhibits sufficient mechanical properties due to the assisted propagation of cracks along the thick α-colonies at the grain boundary. However, no significant difference between α-colony structures of the three different walls could be observed.

Black and white bands that are observed on each wall microstructure (Figure 12) are heat affected zone (HAZ) bands. Ho et al. reported that the dark etching region with an increased α-lamellar spacing, which corresponds to the lower temperature HAZ, developed due to coarsening and greater chemical partitioning [44]. It was also reported that a thin white band occurs on re-heating to just below the β-transus temperature and is associated with a morphological change to the fine α-lamellar colony morphology [44]. The band morphology does not significantly vary between the three manufactured walls, DCEN wall HAZ bands appear to have a greater horizontal curvature due to geometrical inconsistency during printing. There are less HAZ bands in the DCEN wall simply because there were two less thermal cycles compared to the CMT or CSC walls manufacturing.

In summary, it can be noted that there is no critical difference in the printed walls microstructure, thus the metal transfer at the investigated intervals of the transfer parameters primarily influences the geometrical properties of the manufactured structures. Therefore, although the other works on Ti-WAAM are generally based on the CMT transfer process [26,36,37], the CSC transfer appears to be the most suitable for thin-walled structure printing due to its width efficiency and sufficient arc stability.

## 4. Conclusions

In this study, three metal transfer modes were applied in GMAW-based low current Ti-WAAM, namely CMT, CSC, and DCEN. The transfer modes were investigated, results of their application were compared, and the conclusions are as follows:During CMT, the burning arc at the ignition start misses the wire tip, which occurs almost in every cycle and can significantly affect the heat input distribution. Such a phenomenon can be explained by the cumulative effect of the low wire tip temperature and the refractory properties of Ti, which results in insufficient anode vaporization; the lack of anode vaporization develops in the diffusive arc burning mode until the wire tip is heated enough to produce the anode spot. The discovered effect tends to play a crucial role in arc burning stability during CMT.The effective wall width considerably differs between walls, printed using different transfer modes. The width efficiencies are 0.62, 0.76, and 0.56 for the CMT, CSC, and DCEN walls, respectively. The wall width efficiency is influenced by the wall waviness, which depends on the heat input distribution. Heat input distribution varies during GMAW due to arc wandering, and its effect is greater in the CMT process as a result of the extended arc length. The DCEN wall inconsistency is a consequence of the higher heat input into the anode, which in the case of DCEN is the wall. The relatively high CSC wall efficiency is the result of the low arc length and sufficient heat input, produced during the arc burning phase of the metal transfer cycle.There was no critical difference found in the printed walls microstructure, thus the metal transfer at the investigated intervals of the transfer parameters primarily influences the geometrical properties of the manufactured structures.Owing to its width efficiency and sufficient arc stability, the CSC transfer process was found to be the most suitable for thin-walled structure printing among the three studied metal transfer modes. The next study will be dedicated to further development of CSC metal transfer at higher process rates.

## Figures and Tables

**Figure 1 materials-14-02457-f001:**
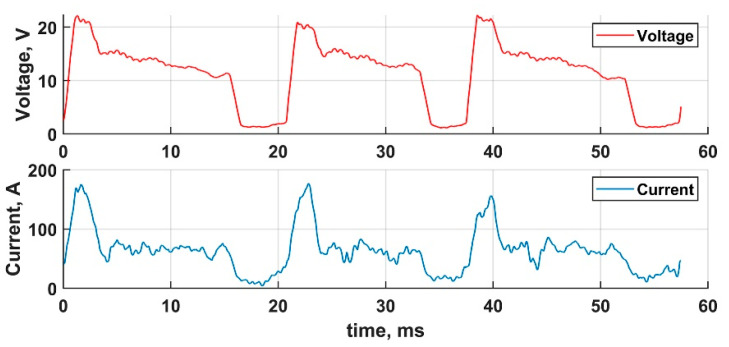
Three definitive CMT cycles.

**Figure 2 materials-14-02457-f002:**
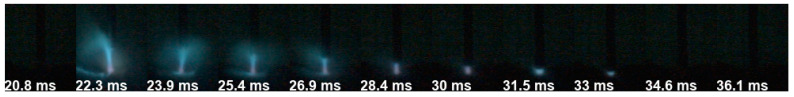
CMT cycle, time correlates with the waveforms in Figure 1.

**Figure 3 materials-14-02457-f003:**
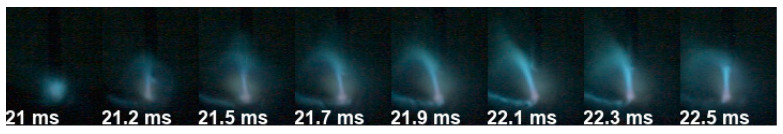
CMT ignition, time correlates with the waveforms in Figure 1.

**Figure 4 materials-14-02457-f004:**
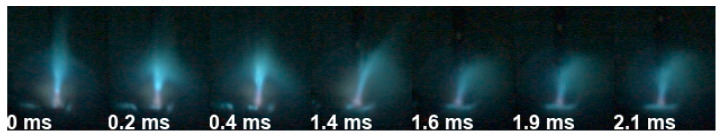
CMT rare patter, occurred at the start of the ignition.

**Figure 5 materials-14-02457-f005:**
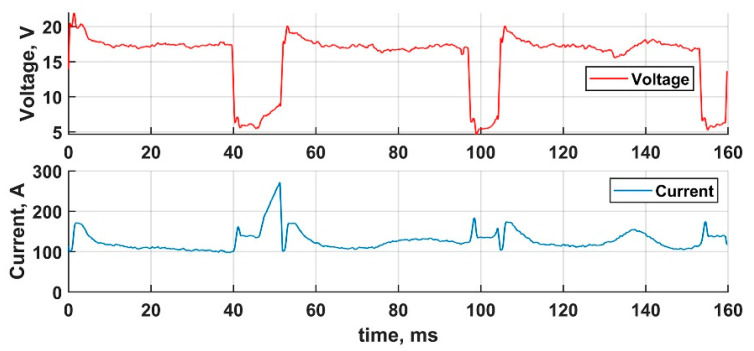
Three definitive CSC cycles.

**Figure 6 materials-14-02457-f006:**
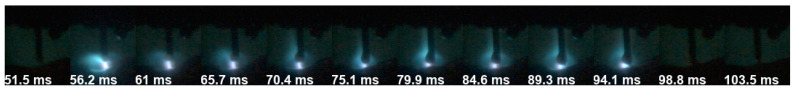
CSC cycle, time correlates with the waveforms in Figure 5.

**Figure 7 materials-14-02457-f007:**
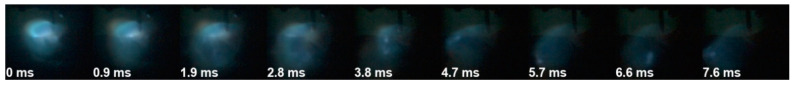
CSC cathode spot wandering.

**Figure 8 materials-14-02457-f008:**
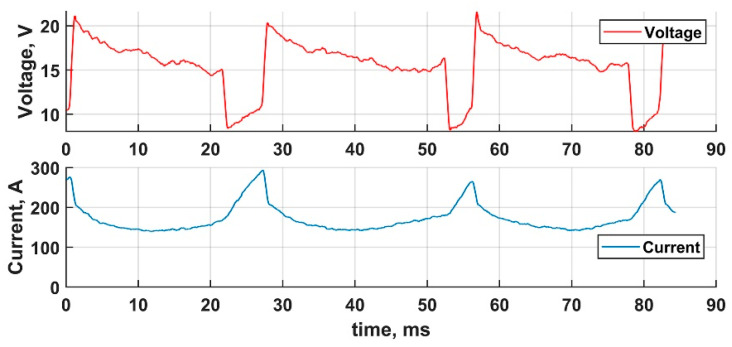
Three definitive DCEN cycles.

**Figure 9 materials-14-02457-f009:**
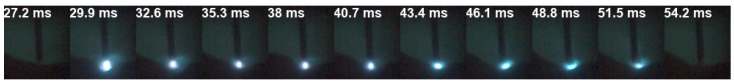
DCEN cycle, time correlates with the waveforms in Figure 8.

**Figure 10 materials-14-02457-f010:**
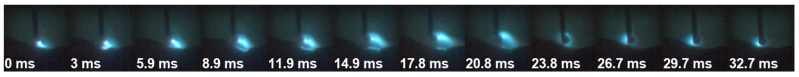
DCEN cathode spot wandering.

**Figure 11 materials-14-02457-f011:**
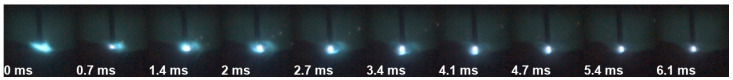
DCEN spattering.

**Figure 12 materials-14-02457-f012:**
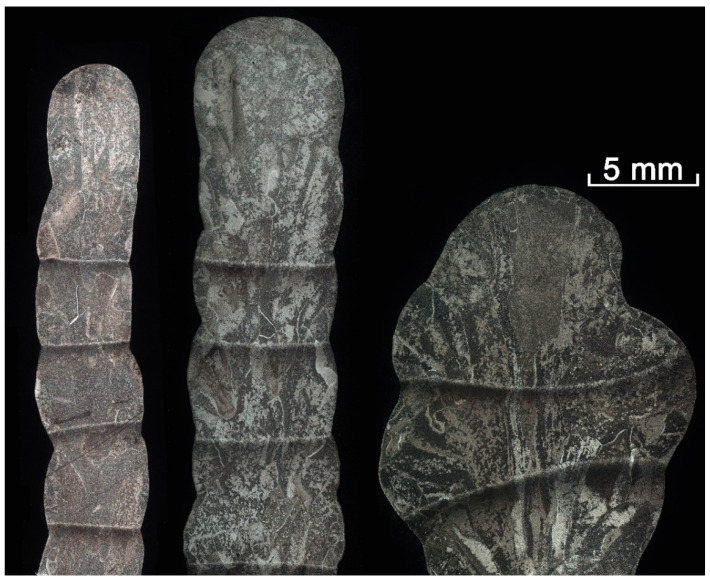
Walls: CMT—on the left, CSC—in the center, DCEN—on the right.

**Figure 13 materials-14-02457-f013:**
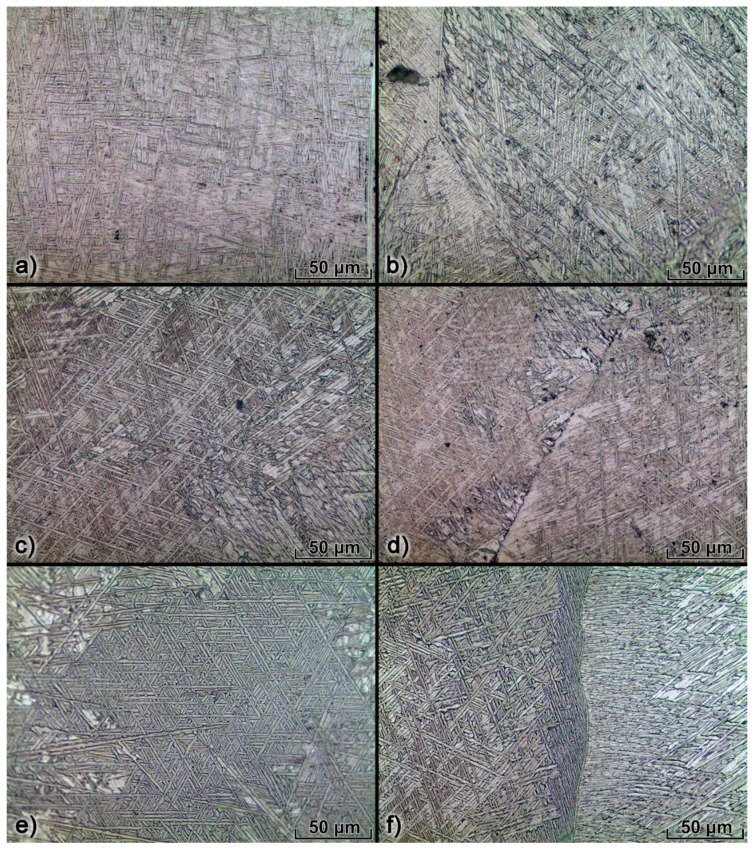
Each wall microstructure: (**a**) CMT α-basketweave in primary-β, (**b**) CMT α-colony along grain boundary, (**c**) CSC α-basketweave in primary-β, (**d**) CSC α-colony along grain boundary, (**e**) DCEN α-basketweave in primary-β, (**f**) DCEN α-colony along grain boundary.

**Table 1 materials-14-02457-t001:** Average current and average voltage values during CMT, CSC, and DCEN transfer processes.

Metal Transfer Mode	Average Current, A	Average Voltage, V
CMT	63	11.3
CSC	127	15.7
DCEN	174	15.6

## Data Availability

The data presented in this study are openly available in Mendeley Data at http://dx.doi.org/10.17632/8ck8h6z3wp.1 (accessed on 13 April 2021), reference number [40].

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
