# Peer review of "Gas Metal Arc Welding Modes in Wire Arc Additive Manufacturing of Ti-6Al-4V"

_materials, 2021, doi:10.3390/ma14092457_

Round 1

Reviewer 1 Report

The work presented for review concerns mainly the transformation of the material into WAAM from Titanium.
The paper discusses the current methods of welding with the methods of additive manufacturing with the processes of arc welding in the gas shield.
I believe that the presented research results, their analysis and conclusions are correct.The article itself contributes to broadening the knowledge in the field of additive arc production. From the editorial side, the article is well written. The drawings are legible and form of presentation of the results is correct.
In my opinion, the work is suitable for publication in Materials

Author Response

Reviewer #1: The work presented for review concerns mainly the transformation of the material into WAAM from Titanium.

The paper discusses the current methods of welding with the methods of additive manufacturing with the processes of arc welding in the gas shield.

I believe that the presented research results, their analysis, and conclusions are correct. The article itself contributes to broadening the knowledge in the field of additive arc production. From the editorial side, the article is well written. The drawings are legible and form of presentation of the results is correct.

In my opinion, the work is suitable for publication in Materials.

Reviewer #1 comment (checkbox) (1). Does the introduction provide sufficient background and include all relevant references? - Can be improved.

Response (1): The introduction section was improved: the choice of the Ti-6Al-4V was explained, one more relevant research (Choudhury et al.) was discussed.

Reviewer #1 comment (checkbox) (2). Are the conclusions supported by the results? - Can be improved.

Response (2): The conclusions section was improved: information on the possible practical application of the research results was presented, as well as it was discussed in the discussion section.

Reviewer 2 Report

Dear aothors,

The presented work on GMA welding and additive manufacturing of Titanium is scientific sound and of interest to the reader. The work focusses on the material transition in WAAM of Titanium. The chosen three different welding modes are the state of the art for additive manufacturing with gas metal arc welding processes. The work summarizes and completes the work on the material transition of titanium alloys well. Furthermore, it contributes to a wider knowledge in arc additive manufacturing. The introduction provides sufficient background and intention for the presented studies. Some of the newest literature like: https://www.sciencedirect.com/science/article/abs/pii/S1526612521001997
is missing, but this has no influence on the quality of the presented work.

The paper is well written and easy to understand.

The used Materials and Methods are well described and provide all necessary information.

The presentation of the results are straight forward and sufficient.

The derived findings are supported by the results.
The conclusions are brief but supported by the results and adress the main focus of the work.

Overall the presented work is in my opinion suitable for publication.

Best regards 

a reviewer

Author Response

Reviewer #2: The presented work on GMA welding and additive manufacturing of Titanium is scientific sound and of interest to the reader. The work focusses on the material transition in WAAM of Titanium. The chosen three different welding modes are the state of the art for additive manufacturing with gas metal arc welding processes. The work summarizes and completes the work on the material transition of titanium alloys well. Furthermore, it contributes to a wider knowledge in arc additive manufacturing. The introduction provides sufficient background and intention for the presented studies.

The paper is well written and easy to understand.

The used Materials and Methods are well described and provide all necessary information.

The presentation of the results are straight forward and sufficient.

The derived findings are supported by the results.

The conclusions are brief but supported by the results and address the main focus of the work.

Overall the presented work is in my opinion suitable for publication.

Best regards

a reviewer

Reviewer #2 comment. Some of the newest literature like: https://www.sciencedirect.com/science/article/abs/pii/S1526612521001997

is missing, but this has no influence on the quality of the presented work.

Response: The suggested relevant research was observed, the results of this study were mentioned in the introduction section and compared to the results of our research in the discussion section.

Reviewer 3 Report

In the paper are presented a series of information on the use of gas metal arc welding (GMAW) for the additive manufacture of parts of Ti alloys.

From the analysis of the information presented in the article, I found the following:

- The paper presents a series of results that may be of interest to the scientific community:

-Introduction needs to be improved, in the sense that references to bibliographic sources such as [1-16], [17-22] are not accepted. Greater emphasis should also be placed on the analysis of publications dealing with Ti alloys and not as in the case of the paragraph of lines 54-58, which refers to aluminum alloys;

- The properties of the Ti alloy used and the reasons behind the choice of this type of material in the research must be presented. It is also necessary to present the parameters of the technological process (voltage, currents);

- The presented images have a low resolution, especially the metallographic structures. The sample preparation technology should be presented in more detail;

- The discussion part needs to be completed, so as to highlight the novelty brought by research compared to other research in the field;

- The conclusions should include a series of information on the practical possibility of using the research presented, as well as future research directions.

Author Response

Reviewer #3: In the paper are presented a series of information on the use of gas metal arc welding (GMAW) for the additive manufacture of parts of Ti alloys.

From the analysis of the information presented in the article, I found the following:

- The paper presents a series of results that may be of interest to the scientific community:

Reviewer #3 comment (1). Introduction needs to be improved, in the sense that references to bibliographic sources such as [1-16], [17-22] are not accepted. Greater emphasis should also be placed on the analysis of publications dealing with Ti alloys and not as in the case of the paragraph of lines 54-58, which refers to aluminum alloys.

Response (1): Bibliographic references were reconsidered where there were insufficiently many of those.

References to WAAM with other alloying systems represent the fact that GMAW is generally used when dealing with steels or Al-alloys due to significant benefits of GMAW. The paragraph in the introduction section, which explains the most important advantages of GMAW is now clarified: there is now no emphasis on the alloying system used, but the GMAW benefits in comparison with GTAW or PAW are now highlighted.

Reviewer #3 comment (2). The properties of the Ti alloy used and the reasons behind the choice of this type of material in the research must be presented. It is also necessary to present the parameters of the technological process (voltage, currents).

Response (2): A paragraph was added to the beginning of the introduction section, representing the reasons behind the choice of the Ti-6Al-4V alloy.

The average current and voltage parameters are now presented in the form of a table (Table 1) in the Materials and Methods section.

Reviewer #3 comment (3). The presented images have a low resolution, especially the metallographic structures. The sample preparation technology should be presented in more detail.

Response (3): We now provide the uncompressed metallographic structure images with higher resolution (figures 12 and 13).

The sample preparation technology is now explained in detail in the Materials and Methods section.

Reviewer #3 comment (4). The discussion part needs to be completed, so as to highlight the novelty brought by research compared to other research in the field.

Response (4): The novelty of our research is now discussed at the end of the discussion section.

Reviewer #3 comment (5). The conclusions should include a series of information on the practical possibility of using the research presented, as well as future research directions.

Response (5): Information on the practical possibility of using the research results and the future research directions are now pointed in the conclusions section.

Round 2

Reviewer 3 Report

The authors revised their manuscript according to my suggestions. Thus the manuscript can be accepted for publication.